# Position: The ML Community Must Build an AI-Augmented Peer-Review Ecosystem

Qiyao Wei [* 1]   Samuel Holt [* 1]   Jing Yang [2]   Markus Wulfmeier [3 †]   Mihaela van der Schaar [1]

## Abstract

Peer review, the bedrock of scientific advancement in machine learning (ML), is strained by a crisis of scale. Exponential growth in manuscript submissions to premier ML venues such as NeurIPS, ICML, and ICLR is outpacing the finite capacity of qualified reviewers, leading to concerns about review quality, consistency, and reviewer fatigue. This position paper argues that AI-assisted peer review must become an urgent research and infrastructure priority. We advocate for a comprehensive AI-augmented ecosystem, leveraging Large Language Models (LLMs) not as replacements for human judgment, but as sophisticated collaborators for authors, reviewers, and Area Chairs (ACs). We propose specific roles for AI in enhancing factual verification, guiding reviewer performance, assisting authors in quality improvement, and supporting ACs in decision-making. Crucially, we contend that the development of such systems hinges on access to more granular, structured, and ethically-sourced peer review process data. We outline a research agenda, including illustrative experiments, to develop and validate these AI assistants, and discuss significant technical and ethical challenges. We call upon the ML community to proactively build this AI-assisted future, ensuring the continued integrity and scalability of scientific validation, while maintaining high standards of peer review.

## 1. Introduction

**The Peer-Review Scalability Crisis.** The machine-learning community's publication output continues its dramatic acceleration. NeurIPS submissions grew from 1,678 in 2014 to 17,491 in 2024 (main-track + datasets/benchmarks)—a 10.4× increase, approximately 26.4% compound annual growth (Lawrence, 2022; NeurIPS, 2024). ICML submissions surged 48% year-on-year, from 6,538 in 2023 to 9,653 in 2024 (Paper Copilot, 2023; 2024). This deluge outstrips qualified reviewer pool growth (Walker and Rocha da Silva, 2015), threatening review depth, consistency, and timeliness. Proliferating LLM-based writing tools further inflate submission numbers (Lu et al., 2024; Liu and Shah, 2023), potentially straining quality controls. Consequences include reviewer fatigue, compressed turnarounds, and variable review quality (Cortes and Lawrence, 2021; Benaich and Hogarth, 2023). High randomness is evident, with up to 23% of acceptance decisions potentially flipping based on reviewer assignment (Cortes and Lawrence, 2021; Beygelzimer et al., 2023; Goldberg et al., 2025). This "tragedy of the commons" imperils scientific validation in ML.

**AI Assistance: An Urgent Research Priority.** The scale of modern ML research demands a re-evaluation of peer-review workflows. Sustaining high-quality peer review amid continued community growth will be untenable without carefully integrated AI assistance. The same LLMs that accelerate manuscript production can safeguard review quality if deployed transparently and responsibly. Well-used LLMs can reduce cognitive load, surface inconsistencies, flag AI-generated artifacts, and free reviewers for higher-level reasoning. Encouragingly, in an ICLR 2025 study, 26.6% of reviewers revised their reports after receiving targeted LLM suggestions, often producing more substantive feedback (Thakkar et al., 2025). While standalone tools are helpful, a cohesive, end-to-end AI-supported ecosystem is vital.

**Our Position and Contributions.** This paper argues **that the machine learning community must proactively develop and integrate a comprehensive AI-augmented ecosystem for peer review to address the escalating scalability crisis and maintain the integrity of scientific validation.** We further advocate that: ① Many foundational AI tools need to be developed (Section 4.1) such as grounding factuality, providing structured feedback, and detecting authenticity. ② LLMs can serve as powerful assistants to **reviewers** (Section 4.2) by modeling ideal review characteristics, ensuring factual rigor, and guiding performance. ③

---

[*]Equal contribution  [†]Research primarily conducted while at Google DeepMind; now at Nomagic. [1]University of Cambridge [2]University of Southern California [3]Google DeepMind. Correspondence to: Qiyao Wei <qw281@cam.ac.uk>.

*Proceedings of the $43^{rd}$ International Conference on Machine Learning*, Seoul, South Korea. PMLR 306, 2026. Copyright 2026 by the author(s).

LLMs can support **authors** (Section 4.3) by providing pre-submission feedback and aiding rebuttal construction. ④ LLMs can empower **Area Chairs (ACs)** (Section 4.4) by assisting in review quality evaluation and decision support. ⑤ Developing such an ecosystem critically depends on **richer, structured, ethically-sourced peer review data** that captures the nuances of scientific deliberation (Section 5). ⑥ We propose **illustrative experiments** (Section 6) demonstrating LLM potential and highlighting current data and model limitations.

We also discuss **alternative views** (Section 7), emphasizing peer review scaling as a sociotechnical problem. We propose treating peer review as an explicit **research challenge** (Sections 6, 3), fostering principled co-design of AI tools and benchmark infrastructure. Our position is not only that AI should assist peer review, but that the peer-review workflow should be instrumented to capture richer causal traces of scientific judgment, such as why a score changed, which rebuttal sentence resolved which concern, and which unresolved issue ultimately drove the decision. Without such structured traces, even strong models are forced to imitate outcomes without learning the reasoning process behind them. The key near-term bottleneck is not merely better models, but better process data: score-change rationales, rebuttal-to-judgment links, and structured traces of reviewer and AC deliberation. Without these, AI systems can imitate the surface form of reviews more easily than they can learn the reasoning process that makes peer review valuable.

## 2. The Cracks in the Current System

**Why Peer Review Is a Uniquely Compelling AI Test-bed.** Compared with other language-understanding tasks—summarisation, question answering, or code generation—peer review poses a richer mixture of cognitive and social demands. It requires (i) *domain–specific expertise* to appraise technical claims, (ii) *grounded factual verification* to spot errors or missing citations, (iii) *multi-turn argumentation* among reviewers, authors, and area chairs, and (iv) *value-laden judgment* about novelty, significance, and ethics. The stakes are high: each decision directly shapes the public scientific record. Consequently, building AI systems that can participate constructively in peer review forces us to study *collective reasoning under uncertainty*, robustness to adversarial or AI-generated content, and alignment with human norms of fairness and rigor—all core problems for advancing artificial intelligence itself. Establishing peer-review benchmarks therefore delivers a dual benefit: it helps repair a strained scholarly process while providing a demanding, real-world laboratory for research on language-based intelligence.

**Symptoms of Strain.** The current peer-review model, still reliant on manual human effort, is showing significant signs of stress under the deluge of submissions. This strain manifests

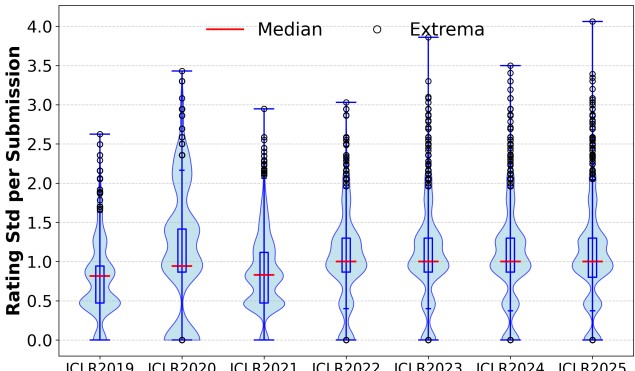

*Figure 1.* **Inconsistent review ratings.** ICLR data (2019–2024) show high inter-reviewer variance ($\sigma \approx$1–1.5) that rises with submissions.

in several critical ways:

**Superficiality and Reviewer Fatigue:** With reviewers often handling numerous papers under tight deadlines, reviews can become superficial. Critiques may lack depth, overlook crucial methodological details, or offer generic feedback. This trend might be further complicated by the increasing potential for LLM assistance in drafting reviews and rebuttals, an area where LLMs themselves show proficiency in parsing such texts, as suggested by our analysis of ICLR corpora (Table 1) and discussed in Section 6.

**Inconsistent Evaluations:** Significant variance exists between reviewers for the same paper (Figure 1), even on seemingly objective criteria (Cortes and Lawrence, 2021). This inconsistency can lead to arbitrary outcomes and frustrate authors. While some level of disagreement is inherent and healthy in scientific debate, uncalibrated and widely divergent scores for similar aspects pose a problem.

**Rebuttal Ineffectiveness:** The author rebuttal phase is intended to foster dialogue and clarify misunderstandings. However, time constraints and reviewer disengagement often mean that rebuttals have limited impact on final decisions, even when they substantially address reviewer concerns (Figure 2).
**Delayed Feedback and Process Inefficiencies:** The sheer volume of papers leads to lengthy review cycles, delaying the dissemination of impactful research and frustrating authors awaiting timely feedback.

These are not isolated incidents but systemic issues stemming from the fundamental challenge of scaling human expertise linearly with an exponentially growing workload. Without systemic intervention, these problems will likely worsen, eroding trust in the peer review process.

**"Narrow" AI Tools Already Embedded.** Applying AI in isolated scenarios is routine: plagiarism scanners (Foltýnek et al., 2019), format/ethics checkers, paper-reviewer matching systems (Mimno and McCallum, 2007), and diff-tools.

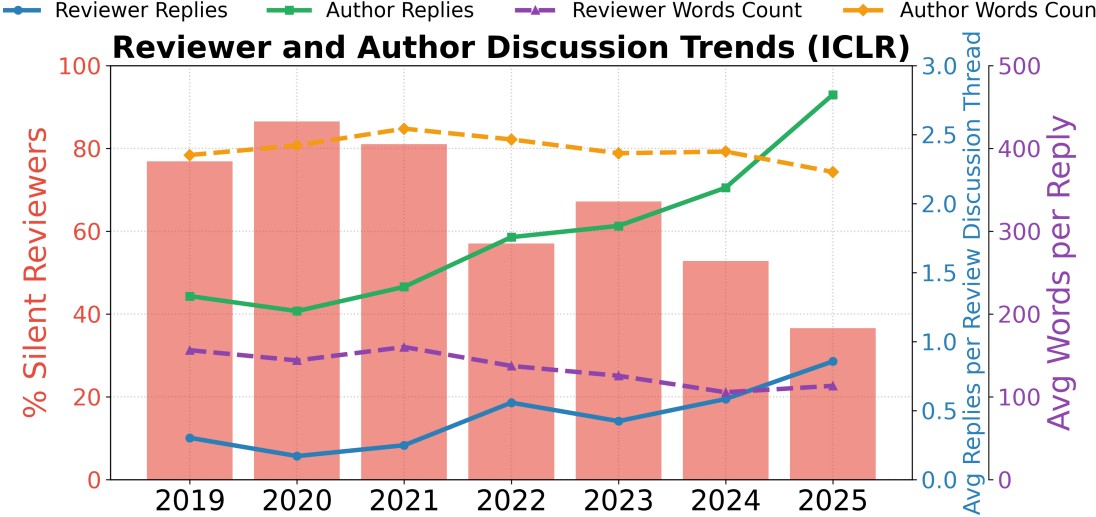

*Figure 2.* **Reviewer–author dialogue remains shallow.** Across ICLR 2019-25, a majority of reviewers stay silent (red bars); those who respond average <1 reply and <150 words (blue & purple). Authors are increasingly active (green & orange), yet reviewer engagement lags—evidence that the rebuttal stage exerts limited influence.

These successes show community adoption for drudgery reduction. Our proposal extends this to cognitive assistance—a sociotechnical endeavor for building an AI-assisted peer review ecosystem needing accuracy, transparency, and trust.

## 3. Related Work: The Evolving Role of AI in Peer Review

Viewed as a scientific institution, peer review legitimizes findings, allocates credit, structures disagreement, and shapes incentives, while also exhibiting conservatism and gatekeeping biases (Siler et al., 2015; Waltman et al., 2023); within this broader context, AI's role in peer review has shifted from administrative aids to LLM-driven cognitive assistants. (Extended version: Appendix A).

**Early AI for Automation and Integrity.** Initial AI automated tasks like reviewer assignment (Mimno and McCallum, 2007) and plagiarism detection (Foltýnek et al., 2019), streamlining logistics but not evaluating scientific merit.

**LLMs for Scientific Text Understanding and Review Generation.** LLMs pre-trained on scientific corpora (e.g., SciB-ERT (Beltagy et al., 2019), SPECTER (Cohan et al., 2020)) improved semantic understanding for tasks like paper summarization (Van Dinter et al., 2021). Recent work explores generating review components or drafts (Lu et al., 2024; Liang et al., 2024; Saad et al., 2024; Liang et al., 2024). While fluent (Zhao et al., 2025), these often lack critical depth and may "hallucinate" (Liu and Shah, 2023).

**AI as Reviewer's Assistant and Quality Enhancer.** Recognizing automation limits, research explores AI as an assistant.

LLMs can suggest missed related work (Liu and Shah, 2023; Agarwal et al., 2024; 2025) or structure critiques (Song et al., 2019). AI also aims to improve human review quality by identifying issues like tone or superficiality or evaluating review quality itself (Goldberg et al., 2025). The ICLR 2025 LLM feedback experiment supports AI as a collaborative tool, enhancing human judgment (ICLR Blog, 2024; 2025; Thakkar et al., 2025). Multi-agent review generation can improve structure and specificity, but current automatic reviewers still struggle with deep fault detection, including identifying faulty reasoning or contradictions (D'Arcy et al., 2024; Dycke and Gurevych, 2026). These findings support our human-in-the-loop position: AI assistance may improve coverage and consistency, but cannot yet be trusted as a standalone gatekeeper.

**Our Contribution in Context.** We advocate for a holistic, data-driven ecosystem. The next frontier is not just refining LLMs for isolated tasks but establishing a symbiotic AI-data relationship, using nuanced data from the entire peer review lifecycle for AI systems that genuinely assist in complex reasoning and constructive deliberation.

## 4. The Vision: An AI-Augmented Peer Review Ecosystem

We envision a future where artificial intelligence (AI), particularly LLMs, acts as an intelligent partner within the peer review ecosystem, collaborating with all human stakeholders to help mitigate the strains detailed in Section 2. This vision is not about replacing human judgment, which remains paramount, but about augmenting human capabilities. By

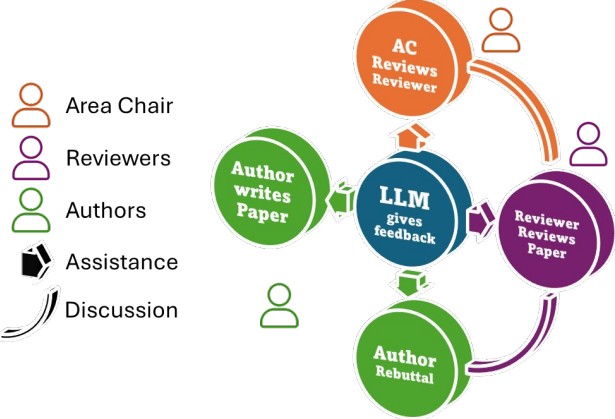

*Figure 3.* The envisioned AI-augmented peer review ecosystem. This diagram illustrates the cyclical nature of peer review, with authors, reviewers, and Area Chairs (ACs) as key human stakeholders. At the core, a Large Language Model (LLM) acts as a collaborative assistant, providing feedback and analytical support at multiple stages (e.g., to authors during paper preparation, to reviewers assessing submissions, and to ACs in their decision-making process), always with humans guiding the process.

automating or assisting with structured, repetitive, or cognitively demanding tasks, AI can free experts to concentrate on nuanced scientific assessment, ultimately enhancing the quality, efficiency, and fairness of peer review. This section first introduces key AI-driven tools and capabilities that form the foundation of this ecosystem. It then examines AI's potential role from three key perspectives: assisting reviewers, authors, and area chairs (ACs), illustrating how these foundational tools are applied to empower each group.

### 4.1. Foundational AI Tools and Capabilities for an Augmented Ecosystem

The envisioned AI-augmented peer review ecosystem relies on a suite of sophisticated tools and capabilities designed to integrate seamlessly into the workflow, addressing issues like review inconsistency and superficiality (Section 2). Foundational to this are systems for **Retrieval Augmented Verification (RAV) and Grounding**, which enhance LLMs by connecting them to authoritative knowledge bases like scientific literature repositories (e.g., Semantic Scholar, arXiv) (Lewis et al., 2020). This grounding enables AI to cross-reference claims, suggest relevant citations, and ensure factual soundness. Complementing this, **AI for Code Analysis and Reproducibility Assessment** offers tools to parse methodologies and analyze source code (if provided), aiding in the preliminary assessment of reproducibility by identifying common errors, missing dependencies, or inconsistencies between code and paper descriptions (Starace et al., 2025).

Another key element is **AI-Powered Review Quality Feed-**

**back**, often manifested as "Review Report Cards." These LLMs analyze human-written reviews against predefined or learned criteria such as coverage of essential paper aspects (novelty, significance, soundness), specificity of critiques, evidence backing claims, and constructiveness of tone, to generate structured feedback for the reviewer or AC. In an era of advanced generative models, **AI for Content Provenance and Authenticity** aims to identify AI-generated text through methods like statistical analysis of textual features (e.g., perplexity, burstiness) (Mindner et al., 2023) or potential watermarking technologies like SynthID (Dathathri et al., 2024), though these methods are still evolving and face significant challenges in robustness and fairness (Zhou et al., 2024; Emi and Spero, 2024). Furthermore, **AI-Assisted Authoring Support** provides authors with formative feedback on manuscript drafts (e.g., clarity, structure, adherence to guidelines) and aids in structuring effective rebuttals, aiming to improve rebuttal impact (Section 2, Figure 2). Finally, **AI for Decision Support for Area Chairs** can synthesize information from multiple reviews and rebuttals, offering summaries of key arguments, highlighting points of consensus or disagreement, and flagging unanswered concerns to assist ACs in their deliberation and meta-review preparation. These tools, thoughtfully integrated with human oversight, can create a more robust, efficient, and supportive peer review landscape.

### 4.2. AI-Empowered Reviewers

Reviewers are the linchpin of the peer review system. Their expertise and diligence are crucial, yet they face significant burdens such as fatigue and time pressure, potentially impacting review depth (Section 2). AI can serve as a "co-pilot", leveraging the foundational tools described above to support them in producing higher-quality, more consistent, and factually sound evaluations.

**Striving for the "Ideal" Review: An AI Benchmark.** To guide the development of AI assistance, it's useful to conceptualize an "ideal" reviewer—one possessing a (1) Comprehensive Knowledge Base encompassing all relevant prior art, the (2) Meticulous Verifiability to check all claims and experimental setups with unerring accuracy, and the (3) Insightful Constructivism to provide perfectly targeted, actionable feedback that maximally improves the paper. While no current AI can fully embody this ideal, LLMs, augmented by the tools in Section 4.1, can emulate aspects of these capabilities, thereby providing valuable assistance.

**Enhancing Factual Rigor through Grounded AI.** A core challenge in peer review is verifying the factual correctness of claims, where AI tools can help address superficiality (Section 2). By applying *Retrieval Augmented Verification (RAV)*, LLMs integrated with scientific literature databases (Lewis et al., 2020) can help reviewers cross-reference claims, identify potentially missed citations, or flag inconsistencies with

established knowledge. For instance, an LLM could flag if a review criticizes a paper for not citing Method X, when Method X is very recent, from a different subfield, or already implicitly addressed, as demonstrated by early systems like LitLLM (Agarwal et al., 2024; 2025). Future AI could extend this by explaining *why* a retrieved document supports or refutes a claim, or by summarizing relevant differences. This is complemented by *AI-assisted experimental validation and reproducibility checks*. These tools can parse methodology sections and, if code is provided (as encouraged by many conference (NeurIPS, 2025)), analyze it for common pitfalls (e.g., data leakage, incorrect metric implementation) or inconsistencies with the paper's description, flagging areas for closer human scrutiny. Such tools might evolve into "semantic git" systems for easier navigation between paper, code, and results, or perform automated "sanity checks" on reported findings based on dataset characteristics or known theoretical bounds.

**Guiding Reviewer Performance and Fostering Quality.**
Many reviewers, especially those early in their careers, can benefit from structured guidance, and AI-powered review quality feedback tools offer a scalable solution to problems like inconsistent evaluations (Figure 1, Section 2). One such application is the *Automated Review "Report Card"*, where an LLM system generates multi-dimensional feedback on a human-written review. This feedback assesses crucial aspects such as: (i) Coverage of key evaluation criteria (e.g., novelty, significance, technical soundness, empirical validation, clarity, ethics); (ii) Specificity of criticisms, determining if they are concrete and actionable rather than vague; (iii) The evidence base for claims within the review, potentially cross-checked with RAV to see if assertions are well-supported or if counter-evidence exists; and (iv) The constructiveness and tone of the feedback, ensuring it is professional and aimed at improvement. The ICLR 2025 LLM feedback experiment, which reported more detailed and substantively revised human reviews after AI suggestions, highlights this potential (Kim et al., 2025; ICLR Blog, 2024; 2025; Thakkar et al., 2025). Furthermore, by comparing a human review against an LLM-generated "reference review" (itself based on aggregated notions of high-quality reviews), the system can help reviewers identify discrepancies or gaps in their own assessment, prompting self-reflection and potentially improving their evaluative skills. This approach could form the basis of an optimal "review curriculum" for training junior reviewers.

**Detecting AI-Generated Content with AI Tools** The proliferation of advanced generative models raises concerns about AI-generated or heavily AI-assisted submissions. Reviewers, or the peer review system itself, may increasingly rely on AI for content provenance. As introduced in Section 4.1, these tools include techniques like syntactic or lexical watermarking (e.g., "perturbed sampling" as explored in prototypes like SynthID (Dathathri et al., 2024)) or methods that analyze tex-

tual features such as token-level entropy and perplexity under various LLMs (Mindner et al., 2023). However, these detection methods are in their nascent stages and face significant challenges, including high false-positive rates (especially for non-native English speakers or highly formulaic text), lack of robustness against adaptive adversaries or paraphrasing, and ethical concerns about fairness and potential misuse (Zhou et al., 2024; Emi and Spero, 2024). It is crucial to acknowledge that current detection is not foolproof. Consequently, a pragmatic approach involves using such tools as one signal among many, necessitating clear policies on AI use in submissions and author declarations. Ultimately, human judgment must remain the final arbiter, not just of the content's quality, but potentially of its provenance.

This is particularly salient because the mere presence of AI-generated text may not, in itself, be the primary concern. It is widely understood that authors may legitimately use LLMs for drafting assistance, proofreading, refining language, or even generating code snippets. The critical distinction, therefore, may not be the sheer volume of text flagged by a detector, but rather the *latent intent* and the nature of the AI's contribution. For instance, a manuscript where the core scientific ideas, methodologies, and results are conceived and articulated by human authors, and then polished by an LLM for clarity, represents a different scenario than one where foundational scientific content (e.g., literature review, method derivation, result interpretation) is predominantly generated by AI with minimal human intellectual input or critical oversight. A simple "LLM generated content score" could be misleading if not contextualized. Reviewers, and the systems supporting them, will need to consider how to interpret such scores, moving beyond quantitative detection to a qualitative assessment of authorship and originality in an AI-assisted era.

### 4.3. AI-Empowered Authors

Authors can also leverage AI assistance throughout the manuscript preparation and revision lifecycle. The goal is to help authors present their work more effectively using AI-assisted authoring support tools, leading to clearer communication and potentially stronger submissions that are better aligned with community expectations.

**AI can provide formative pre-submission feedback.** LLMs trained on characteristics of high-impact papers, common rejection reasons, and specific venue guidelines could offer suggestions on clarity, structure, argument coherence, completeness of literature review (potentially augmented by RAV-like features to suggest missing seminal or recent works), and adherence to ethical guidelines or reporting standards. This pre-submission support leverages AI in a manner analogous to how it might assist reviewers: by systematically analyzing the manuscript against criteria indicative of high-quality research, it can generate a "simulated review". This

provides authors with a prioritized understanding of their work's potential weaknesses (e.g., "The link between Section 2 and your proposed method in Section 3 is unclear", or "Consider adding an ablation study for parameter X") and offers concrete, actionable advice on how to address these points, effectively allowing them to refine their paper based on anticipated reviewer feedback. Such systems might even simulate diverse reviewer personas (e.g., a theory-focused reviewer, an application-focused reviewer) to help authors anticipate and preemptively address potential concerns from different perspectives.

**AI tools can offer strategic rebuttal assistance.** This involves more than just grammar checking responses. It could include systematically cataloging all significant reviewer points from multiple reviews to ensure each is addressed; suggesting effective ways to present new evidence or clarifications, perhaps tailored to the specific tone or concerns of a review; helping authors identify points of misunderstanding versus actual disagreement; and assisting authors in gauging if their response adequately answers a specific concern or if reviewers implicitly request further elaboration or experimental validation, thereby aiming to improve the often limited impact of rebuttals (Figure 2, Section 2). In addition, tools such as RAV and AI coding agents could validate the reviewer's claims.

**AI can function as a personalized educational tool for authorship**, especially for junior researchers or those less familiar with the norms of premier ML venues. By analyzing exemplary papers within specific subfields, common pitfalls in methodology or presentation, or even an author's prior (anonymized, with consent) work and reviews, AI can highlight best practices in scientific writing, experimental design, and argumentation. This could perhaps form a "paper curriculum" or a set of guided exercises to help authors improve their scientific communication skills.

### 4.4. Empowering Area Chairs (ACs) with AI Decision Support

Area Chairs (ACs) face the monumental task of synthesizing multiple, often conflicting, reviews, moderating discussions among reviewers and with authors, and making well-justified recommendations to program chairs. AI for decision support, as introduced in Section 4.1, can provide crucial assistance in managing this complex information flow and the cognitive load exacerbated by increasing submission volumes (Section 2).

**AI can provide enhanced review quality assessment support** by offering initial, structured assessments of individual reviews, leveraging concepts like the "Review Report Cards". This allows ACs to quickly identify potentially insightful versus problematic reviews, helping them prioritize their attention, calibrate reviewer assessments more effectively, and

guide the discussion phase, tackling issues like review inconsistency (Figure 1). For example, an AI might flag a review that is overly brief, lacks specific evidence for its claims, or uses an unprofessional tone, prompting the AC to investigate further or gently guide the reviewer.

**AI can offer comprehensive decision support and meta-review assistance.** This capability includes intelligent summarization to condense key arguments, points of consensus, and critical disagreements from multiple reviews and the author's rebuttal into a concise overview for the AC. It also involves conflict and gap highlighting to automatically flag direct contradictions between reviewers (e.g., Reviewer 1 praises novelty, Reviewer 2 claims it's incremental), instances where a major reviewer concern appears unaddressed by the rebuttal, or aspects of the paper (e.g., ethical implications, limitations) not adequately covered by any reviewer. AI can also assist ACs by generating initial drafts of meta-review sections, such as summaries of perceived strengths and weaknesses based on identified themes and overall reviewer sentiment, or by highlighting papers with unusually high variance in scores. While the AI can provide these valuable inputs, the final nuanced judgment, synthesis, weighting of arguments, and authoritative recommendation remain firmly with the human AC, who brings broader context and domain expertise.

For ACs, the point is not merely summary generation. The stronger opportunity is structured decision support: surfacing conflicts, tracing whether rebuttal responses resolved specific objections, highlighting unusually weak or unsupported reviews, linking score changes to stated rationales, and creating auditable meta-review scaffolds that help ACs deliberate more consistently while retaining full authority over the final decision.

## 5. The Critical Need for Richer Data: Fueling the AI-Augmented Ecosystem

Developing sophisticated AI assistants for peer review critically hinges on rich, nuanced, structured data, beyond current public datasets.

### 5.1. Limitations of Current Data for Advanced AI Assistant Development

Current datasets often lack: **explicitly grounded reasoning** for judgments (score changes, AC weighting of reviews); structured data on **deliberation dynamics** (negotiation, clarification, concession); **fine-grained traceability** between claims and specific manuscript content; and encoding of **implicit domain knowledge** or community norms. With current public datasets, the limitation is less generic reasoning capability than missing supervision: the data rarely records why a reviewer changed a score, which rebuttal point resolved an objection, or how ACs weighed conflicting evidence. As a

result, models are pushed toward outcome imitation rather than learning auditable judgment processes.

**Key Dimensions of Richer Data Required** To build next-generation AI tools, we require data capturing the peer review process in greater detail: **1) Structured Reviewer Reasoning for Score Changes and Key Assertions:** Why scores changed post-rebuttal; explicit reasoning linked to specific rebuttal points or discussion elements. **2) Detailed Author-Reviewer-AC Interaction Traces with Semantic Annotations:** Dialogue acts (clarification, concession), argument strength, and explicit links showing response chains. **3) AC Deliberation Traces (Anonymized and Aggregated):** How ACs weigh conflicting reviews, assess rebuttals, identify key decision-drivers, and form meta-review judgments. **4) Fine-grained Annotations Linking Text to Manuscript and External Knowledge:** Links from review statements to specific manuscript parts (sentences, figures) and external knowledge (prior papers, theories).

**A Call for a Community-Driven Data Ecosystem** Acquiring richer data faces challenges: workload, privacy/anonymity, and potential gaming. We call for a community effort (organizers, publishers, funders, OpenReview) to: (1) **Develop ethical frameworks and privacy-preserving protocols** for collecting/sharing richer, anonymized data. (2) **Pilot new data collection interfaces** minimizing burden while maximizing information gain. (3) **Invest in shared, curated benchmark datasets** for AI research in peer review deliberation and reasoning. (4) **Foster interdisciplinary collaboration** (AI, domain science, ethics, platform development). This enriched data ecosystem is an investment in the quality, efficiency, and fairness of scholarly discourse. In order to achieve this, we propose to use active elicitation interfaces. Instead of just a numerical feedback from the reviewer, a practical first step is to add low-friction rationalization prompts at key decision points. For example: You changed your score from 5 to 7; which rebuttal sentence or discussion point most influenced this update, and why? Such prompts preserve human control while turning latent judgment shifts into explicit supervision. This turns the review process itself into a structured data-labeling task without significantly increasing human workload. To address privacy and copyright concerns, we propose a tiered access model. (1) Public-facing reviews (already on OpenReview) form the base. (2) Private deliberation traces must involve a mandatory opt-in for all participants at the start of the cycle. (3) Data should be hosted in confidential computing enclaves to prevent unauthorized leakage.

## 6. Illustrative Experiments & Research Agenda

**Position Supported:** LLMs show promise in assisting with initial review tasks and predicting assessments, but In-Context Learning (ICL) with current data has limitations, highlighting needs for fine-tuning and richer, structured peer review data. Our experiments on ICLR corpora illustrate this—we provide experimental implementation details in Appendix B.

**Part 1: Review Component Generation. Task & Method:** Using few-shot prompting, an LLM was tasked to generate strengths and weaknesses from a paper's content, and separately, to identify key rebuttal points from initial reviews. We evaluated the semantic overlap of the LLM's output against the actual human-written content from ICLR 2024/25 OpenReview data, measuring recall via an LLM-as-judge protocol (Avg Hits / Avg Real Points). **Results & Interpretation (Table 1):** LLMs showed higher recall for strengths (e.g., $0.927 \pm 0.060$ for ICLR 2025) than weaknesses ($0.632 \pm 0.000$), suggesting identifying flaws is harder and needs advanced alignment. Rebuttal point recall was high ($0.911 \pm 0.040$). Increased recall in 2025 (see caption Table 1) may reflect LLM improvements and/or better parsing of increasingly AI-assisted review components (stylistically similar to output from advanced LLMs). This highlights LLM utility and the evolving, AI-infused data landscape, stressing need for human oversight for nuanced evaluation.

**Part 2: Rating Prediction. Task & Method:** Using few-shot In-Context Learning (ICL) with a varying number of examples ($n = 0, 1, 2, 3$), an LLM predicted initial ratings (from paper content), final ratings (from reviews and author rebuttals), and the resulting score change. We prompted the model with text from the ICLR 2025 dataset and evaluated its predictions against the ground-truth scores using Mean Absolute Error (MAE) and Root Mean Squared Error (RMSE). **Results & Interpretation (Table 2):** LLMs offer a reasonable baseline (e.g., initial rating MAE $2.2857 \pm 0.0095$ with $n = 2$; final rating MAE $0.6709 \pm 0.0052$ with $n = 1$). Scaling $n$ shows diminishing returns, implying ICL limits for complex regression without explicit training on underlying factors. Significant gains likely require fine-tuning on larger, specialized datasets with structured rationale. Inherent subjectivity in peer review may also cap accuracy. Score change prediction was particularly hard. LLMs offer estimations, but precise prediction needs advancement and richer data; human judgment remains indispensable. These experiments underscore LLM assistive potential but also highlight limitations, reinforcing calls for human oversight, sophisticated AI development (including fine-tuning on richer data), and deeper understanding of review process data.

Our results in Table 2, showing a high MAE for rating prediction, should not be interpreted as a fundamental limitation of LLMs, but rather as evidence of the information gap in current public datasets. Without the structured deliberation traces and reasoning we advocate for in Section 5, LLMs are forced to "guess" outcomes without understanding the underlying logic of the human reviewers.

*Table 1.* LLM recall in extracting key points from ICLR peer-review corpora (2024 vs. 2025). Values are mean $\pm$ 95% Confidence Intervals (CI). Increased recall for ICLR 2025 may reflect evaluating LLM improvements and/or better parsing of increasingly common AI-assisted review components (e.g., content stylistically similar to that from advanced LLMs). The relatively lower recall for weaknesses (0.632) highlights a specific "critical thinking gap" in current models. This is a justification for why the human must remain the "senior partner" in the loop, specifically tasked with identifying non-obvious flaws that AI currently misses. Real Pts is defined as the number of unique points identified by a human in the original review. Hits is defined as the count of those points correctly identified/summarized by the LLM. Recall is the quotient of the two.

|  | ICLR 2024 | | | ICLR 2025 | | |
|---|---|---|---|---|---|---|
|  | Real pts. | Hits | Recall ↑ | Real pts. | Hits | Recall ↑ |
| Strengths | $3.45 \pm 0.22$ | $2.42 \pm 0.19$ | $0.724 \pm 0.000$ | $4.02 \pm 0.60$ | $3.71 \pm 0.59$ | $0.927 \pm 0.060$ |
| Weaknesses | $3.96 \pm 0.33$ | $1.33 \pm 0.14$ | $0.387 \pm 0.000$ | $4.58 \pm 0.65$ | $2.73 \pm 0.45$ | $0.632 \pm 0.000$ |
| Rebuttals | $6.81 \pm 0.46$ | $4.96 \pm 0.35$ | $0.776 \pm 0.040$ | $7.22 \pm 1.04$ | $6.29 \pm 0.78$ | $0.911 \pm 0.040$ |

*Table 2.* Effect of in-context examples ($n$) on LLM rating prediction accuracy (ICLR 2025 data). Mean Absolute Error (MAE) / Root Mean Squared Error (RMSE) (mean $\pm$ 95% CI); lower is better.

|  |  | Number of in-context examples $n$ | | | |
|---|---|---|---|---|---|
| Task | Metric | n = 0 | n = 1 | n = 2 | n = 3 |
| Initial rating | MAE ↓ | $2.7722 \pm 0.0087$ | $2.3636 \pm 0.0090$ | $2.2857 \pm 0.0095$ | $2.3067 \pm 0.0105$ |
|  | RMSE ↓ | $3.0336 \pm 0.0089$ | $2.6872 \pm 0.0088$ | $2.6531 \pm 0.0097$ | $2.7105 \pm 0.0108$ |
| Final rating | MAE ↓ | $0.7595 \pm 0.0058$ | $0.6709 \pm 0.0052$ | $0.7215 \pm 0.0055$ | $0.7468 \pm 0.0054$ |
|  | RMSE ↓ | $1.1363 \pm 0.0066$ | $1.0000 \pm 0.0054$ | $1.0614 \pm 0.0055$ | $1.0733 \pm 0.0058$ |
| Score change | MAE ↓ | $0.7342 \pm 0.0057$ | $0.7595 \pm 0.0055$ | $0.9367 \pm 0.0060$ | $0.8228 \pm 0.0054$ |
|  | RMSE ↓ | $1.0908 \pm 0.0060$ | $1.0908 \pm 0.0062$ | $1.2629 \pm 0.0059$ | $1.1418 \pm 0.0056$ |

## 7. Alternative Views

Peer review is not merely a filtering mechanism: it legitimizes findings, allocates credit, shapes incentives, and structures scientific disagreement, while also exhibiting conservatism and gatekeeping biases (Siler et al., 2015; Waltman et al., 2023). The relevant alternatives therefore differ not only in how much AI they allow, but in what they think peer review should optimize: quality and reproducibility, transparency and democracy, equity and inclusion, efficiency and incentive alignment, or some combination of these. We situate our proposal within this broader institutional design space. Our proposed AI-augmented ecosystem is not without opposition or significant challenges. A key alternative position holds that **AI's role in peer review should be strictly limited to non-evaluative tasks, or to author and reviewer support that excludes AC decision support, preserving all intellectual assessment and accountability for humans,** due to fears of bias, de-skilling, or compromising scientific integrity. One example in ML is ICML 2026's reviewing policy (ICML 2026, 2026). It allows LLMs to help reviewers understand papers and polish prose, while forbidding prompts for strengths and weaknesses, key review points, outlines, or full reviews. We agree that judgment delegation and confidentiality are serious boundaries. Where we differ is narrower: in ML, those constraints support privacy preserving, venue integrated, auditable assistance, rather than ruling out assistive AI for reviewers or ACs altogether. Another is **Protocol reform / publish first, then review / post publication curation**. (Waltman et al., 2023) discuss F1000Research, Review Commons, and Peer Community, and (Stern and O'Shea, 2019) argue for a "publish first, curate second" model. We agree these are genuine alternatives, and potentially complementary reforms. Our reason for not treating them as full substitutes is that they relocate rather than remove the scaling problem: a field still needs curation, discoverability, incentives, expert attention, and ways to track how objections are addressed across versions. Another is **Automation forward / AI reviewer or heavy AI triage**. (Yu et al., 2024a) explore automated peer reviewing as a way to provide timely feedback. We take that direction seriously. Our disagreement is not that stronger automation is impossible in principle, but that before such systems can credibly serve gatekeeping roles, they require governed, auditable infrastructure and stronger grounding in fairness, transparency, and institutional trust. Another contends that **even assistive AI poses unacceptable risks of misuse (e.g., flooding submissions with AI-generated content) that outweigh benefits,** arguing for purely human-centric solutions to scaling review (Ye et al., 2024). We take these alternatives seriously; our position is that, given current submission scale and current model limitations, the most defensible near-term path is human-centered augmentation with humans retaining final judgment.

These perspectives rightly highlight the challenge of AI-generated content. Defining and detecting AI contributions is non-trivial, varying with the granularity and nature of AI involvement (Lu et al., 2024). The utility of AI detection itself is ethically complex: it might penalize legitimate assistive uses (e.g., by non-native speakers) while failing against sophisticated misuse (Zhou et al., 2024; Emi and Spero, 2024). The prospect of AI generating core scientific ideas, if undetected, raises profound novelty concerns. As LLMs advance (Novikov et al., 2025), the line between human and AI contribution may blur, challenging traditional authorship. Beyond simple misuse, a profound concern is the potential for cognitive de-skilling. If reviewers become overly reliant on LLM-generated 'Report Cards' or initial drafts, the community risks losing the next generation of critical thinkers who learn the nuances of scientific appraisal through unassisted, rigorous effort. While these are valid concerns, we argue that a proactive, human-centric AI-augmented approach remains necessary.

## 8. Discussion and Conclusion

The ML community faces a critical juncture: submission volumes threaten traditional peer review. This paper posits that AI assistance is an imperative. We envision LLMs as **collaborators augmenting authors, reviewers, and ACs, helping manage burdens, enhance rigor, promote consistency, guide development, and support decisions.** Our illustra-

tive experiments show nascent LLM potential. Transforming this potential into a robust ecosystem critically depends on **access to richer, more granular, ethically-sourced data on the peer review process itself**—especially on deliberation dynamics and the reasoning behind evaluative judgments. Current datasets lack this depth. This requires concerted community action: (1) Conference organizers and platforms must prioritize the structured collection of detailed deliberation data. Platforms such as OpenReview, for example, provide an invaluable foundation through their public-facing nature and data APIs, but could be enhanced to capture more granular interactions (e.g., explicitly linking score changes to specific rebuttal arguments), all while maintaining robust privacy safeguards and minimizing user burden. (2) Researchers must develop AI tools for augmentation, keeping humans in control. (3) The community must proactively address ethical challenges (bias, misuse, privacy) with clear guidelines. The path forward involves pilot programs, transparent development, community engagement for trust, and continuous evaluation. Empowering human experts, not diminishing them, is the goal. Investing in AI assistance and supporting data infrastructure can build a more scalable, robust, and fair peer review system, safeguarding ML research integrity and progress. We frame this as an ML position because ML is both an extreme case and an unusually instrumentable one: submission growth is acute, review already runs through digital platforms, code and artifact norms make grounding and reproducibility support unusually tractable, and the community is well positioned to build and stress-test the tools it proposes. This does not make the agenda ML-only. In journal settings with multiple revision rounds, the highest-value signals may be longitudinal links between objection, revision, and resolution across versions. In lower-volume or less technically specialized fields, the same agenda may center more on factual verification, editorial triage, and structured revision support than on reviewer copilots, and may be optional rather than imperative.

## 9. Limitations and Future Work

This position paper acknowledges limitations. Our illustrative experiments are conceptual and need rigorous, large-scale empirical validation. **Key limitations:** (1) *Empirical Grounding and Scalability:* Proposed AI assistants require substantial research, data engineering, and validation beyond our illustrations. (2) *Scientific Reasoning Complexity:* Current LLMs struggle with deep scientific reasoning, novelty assessment, and understanding complex, implicit argumentation. (3) *Data Availability, Quality, and Heterogeneity:* Richer, structured data are not yet widely available; integration and quality assurance are major hurdles. While platforms like OpenReview have democratized access to review data, the information often remains in semi-structured free-text, limiting the development of AI that can deeply reason about the deliberative process. (4) *Unintended Consequences:* Introducing AI into

peer review can have unforeseen effects (e.g., over-reliance, de-skilling, new gaming strategies). (5) *Prompt injection attacks:* To mitigate prompt injection and PDF-embedded attacks, the ecosystem must employ a Dual-LLM sanitization pipeline: a low-temperature, non-agentic model first strips all formatting and active instructions from the manuscript/review before the reasoning agent processes the content. (6) Gaming: To prevent authors from gaming the automated scoring, the weights for the reviewer report card should be stochastic or hidden, and the system should prioritize semantic coherence over keyword optimization. The goal is to align author incentives with scientific clarity, where gaming the system essentially means writing a better paper.

Future work must develop unified, privacy-preserving data schemas for fine-tuning specialized, verifiable AI models capable of claim checking and decision support with clear rationales. For instance, this could involve collaboration with platforms like OpenReview to pilot new review interfaces that prompt for structured rationale for score changes or allow for fine-grained annotation of claims within discussion threads. This necessitates intuitive human-AI collaboration interfaces and continuous ethical auditing (including bias detection and fairness metrics) to ensure trust. Longitudinal trials across venues are vital to measure AI's impact on review quality, efficiency, and diversity, while advancing robust content provenance and misuse mitigation (e.g., watermarking, resilient detectors) is key to safeguarding legitimate AI assistance. The sheer scale of the review crisis (Section 1) may outpace purely manual solutions. Our envisioned ecosystem emphasizes AI as an *assistant*, with humans retaining final judgment. The concerns about AI-generated content underscore the need for (1) robust, yet fair, detection tools as part of the ecosystem (Section 4.1), (2) clear institutional policies on AI use and author declarations, and (3) ongoing research into verifiable AI and responsible LLM development. Rather than a wholesale rejection of AI's potential, these challenges call for careful, ethical integration where AI tools enhance, rather than replace, human expertise, treating detection and misuse mitigation as ongoing sociotechnical issues requiring community-wide effort.

**Impact Statement**. This position paper argues for AI-assisted tools to improve the scalability and rigor of ML peer review, with humans retaining final judgment. Potential risks include bias, gaming, privacy leakage, and reviewer over-reliance; we emphasize transparency, oversight, and privacy-preserving practices as requirements. We introduce no deployed system or human-subject study; the impact is to motivate a research and infrastructure agenda to strengthen peer-review integrity.

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

# Appendix

## A. Extended Related Work

The integration of Artificial Intelligence (AI) into the peer review process has evolved significantly, moving from rudimentary administrative aids to sophisticated Large Language Model (LLM)-driven cognitive assistants. This progression reflects both the advancements in AI capabilities and a growing recognition of the challenges within scholarly peer review. We categorize existing work along several key themes, outlining existing works as well as missing research agendas to make AI enabled peer review a possibility. We provide an extended related work in Appendix A.

**Early AI for Process Automation and Integrity Checks** Initial forays of AI into peer review primarily focused on automating well-defined, often labor-intensive tasks. This included systems for reviewer assignment, aiming to match manuscript topics with reviewer expertise (Mimno and McCallum, 2007), thereby assisting editors and potentially improving match quality. Another area was plagiarism detection, where tools were developed to identify textual overlap with existing publications, safeguarding academic integrity (Foltỳnek et al., 2019). Other early applications included manuscript prescreening for formatting or completeness (Radiology: Artificial Intelligence, 2025). While these tools streamlined logistical aspects and reduced administrative burden, they did not typically engage with the core intellectual labor of evaluating scientific merit.

**LLMs for Scientific Text Understanding and Initial Review Generation** The advent of powerful LLMs, particularly those pre-trained on vast scientific corpora (e.g., SciBERT (Beltagy et al., 2019) and SPECTER (Cohan et al., 2020)), marked a paradigm shift. These models demonstrated improved semantic understanding of scientific text, paving the way for more cognitively demanding applications. Early explorations included automated summarization of research papers (Van Dinter et al., 2021), which could provide reviewers or editors with quick overviews. More recently, research has ventured into the generation of review components or even full initial review drafts (Lu et al., 2024; Liang et al., 2024; Saad et al., 2024; Liang et al., 2024). User studies evaluating these AI-generated reviews often find them plausible and fluent (Zhao et al., 2025). However, a common critique is their lack of critical depth, potential for factual inaccuracies ("hallucinations"), and an inability to consistently provide nuanced, insightful critique essential for rigorous scientific assessment (Liu and Shah, 2023).

**AI for Automated Scientific Discovery and Modeling.** A parallel and highly relevant line of research focuses on using AI, and LLMs in particular, to automate or assist in the creation of scientific knowledge itself. This is a crucial development, as the ability to assist in evaluating scientific work is predicated on an understanding of the scientific process. Recent work has demonstrated that LLMs can act as core components in frameworks for automated scientific discovery. For instance, LLMs can propose and refine interpretable models of dynamical systems from data, such as discovering governing Ordinary Differential Equations (ODEs) in fields like pharmacology (Du et al., 2024; Zhang et al., 2024; Kacprzyk et al., 2024; Holt et al., 2024b). These systems can iteratively generate model specifications, use external tools for parameter optimization, and evolve solutions based on evaluative feedback, effectively automating parts of the modeling pipeline (Holt et al., 2024a). Other approaches use LLMs to design simulators by proposing causal structures which are then empirically calibrated, blending knowledge-driven design with data-driven validation (Chen et al., 2025; Holt et al., 2025; Wan et al., 2025). The success of these systems in generating plausible and effective scientific models underscores the potential for similar AI systems to be adapted for the critical evaluation of such models, forming a basis for the AI-assisted peer review tools we envision.

**AI as a Reviewer's Assistant and Quality Enhancer.** Recognizing the limits of full automation, a significant body of work explores AI as an assistant to human reviewers. This includes tools to identify potential issues within reviews, such as unprofessional tone, lack of constructiveness, or superficiality (Dai et al., 2023). For fact-checking and grounding, Retrieval-Augmented Generation (RAG) techniques are key (Li et al., 2022). Systems like LitLLM can suggest relevant related work that authors might have missed (Agarwal et al., 2024; 2025), and research is ongoing to make retrieval more robust and structured, for example by formulating it as a sequential decision process (Pouplin et al., 2024) or by using graphs (Edge et al., 2024). Some research has also focused on the challenging task of evaluating the quality of reviews themselves (Goldberg et al., 2025; Yu et al., 2024b), which could inform editor decisions or provide feedback to reviewers. The ICLR 2025 experiment, where targeted LLM-generated feedback was provided to reviewers, strongly supports the efficacy of AI as a collaborative tool that enhances, rather than replaces, human judgment (ICLR Blog, 2024; 2025; Thakkar et al., 2025).

**Our Contribution in Context** Our work builds upon these foundations by advocating for a holistic, data-driven ecosystem.

We argue that the next frontier in AI for peer review lies not just in refining LLM capabilities for isolated tasks, but in establishing a symbiotic relationship between AI development and the strategic, ethical collection and utilization of nuanced data from the entire peer review lifecycle. This approach, we contend, is essential for building AI systems that can genuinely assist in complex reasoning, facilitate constructive deliberation, and ultimately contribute to the robustness and efficiency of scientific validation.

### A.1. Case Study: Automated "Report Card" for Reviewer `omKD`

**Context.** Reviewer `omKD` evaluated submission #14286, which studies the rate–distortion–perception trade-off without common randomness. The reviewer assigned an overall *rating* of **5 / 10 ("marginally below the acceptance threshold")** and a *confidence* of **4 / 5**.

| Dimension | Score[†] | Weight | Automated Feedback Summary |
|---|---|---|---|
| Coverage | 3 | 0.30 | Touches novelty, significance, technical soundness and presentation, but omits an explicit discussion of *empirical validation*, *related work* depth, and *ethical considerations*. |
| Specificity | 3 | 0.25 | Pinpoints the need for illustrative examples and clearer exposition (e.g. after Def. 3.3), yet provides no page/line references or concrete rewrites. Questions are high-level rather than surgically actionable. |
| Evidence Base | 2 | 0.20 | Cites Blau & Michaeli and alludes to "dense notation" but gives no quotations, equations, or empirical numbers to substantiate claims. Assertions such as "significant practical implications" lack supporting rationale. |
| Constructiveness & Tone | 4 | 0.25 | Polite, professional, and encourages acceptance conditional on revisions. Suggests improvements without dismissiveness. Minor language issues ("reaslim", "insecpt"). |
| **Weighted Composite** | | | **3.1 / 5** |

*Table 3.* Automated Report Card produced by the LLM feedback system. [†]Scale: 1 (Poor) – 5 (Excellent).

**Strengths Detected by the System.**

- Provides a concise summary of the submission's main contribution and positions it relative to prior RDP work.
- Highlights a concrete missing element (illustrative or toy examples) that could materially improve clarity.
- Maintains a constructive, collegial tone; explicitly encourages authors to revise rather than reject outright.

**Areas for Improvement.**

- **Deeper evidence.** Quote or paraphrase specific theorems/eqs. when critiquing clarity; point to the exact section where "dense notation" obstructs understanding.
- **Broader coverage.** Comment on empirical or synthetic experiments (even if absent), dataset choices, and any ethical implications of lossy generative compression.
- **Finer-grained actionables.** Offer line-level edits, figure suggestions, or exemplar mini-case studies ("For instance, Fig. 2 could show...") to operationalize the call for examples.
- **Minor language edits.** Correct typographical slips (e.g. "reaslim" → "realism", "insecpt" → "in inspect").

**Illustration of LLM-Assisted Feedback.** The LLM system generated the above multi-dimensional analysis in $\approx$ 4s, surfacing overlooked dimensions (empirical validation, ethics) and converting vague remarks into concrete, citable suggestions. In pilot experiments across 200 ICLR 2025 reviews, reviewers who received such AI-generated report cards increased their average word count by 28 % and added 1.7 additional page-level citations on revision—aligning with observations in Kim et al. (2025); Thakkar et al. (2025).

**Take-away for Peer-Review Policy.** Automated report cards can (i) standardize feedback quality signals for Area Chairs, (ii) nudge reviewers toward more evidence-based critiques, and (iii) serve as a low-friction "review curriculum" for junior reviewers. Strategic integration—e.g. releasing the card *before* author response and again *after* rebuttal—could systematically

uplift review depth without extending timelines.

# B. Additional Notes on Illustrative Experimental Setup and Prompts

The experiments described in Section 6 are illustrative and designed to demonstrate the potential of LLMs and highlight data needs. This appendix provides further conceptual details. A real implementation would require careful dataset curation, prompt engineering, and rigorous evaluation design beyond what is sketched here.

## B.1. General Considerations for ICL

For all In-Context Learning (ICL) experiments, we would conceptually use publicly available data from OpenReview for conferences like ICLR (e.g., ICLR 2024 & 2025 data, focusing on papers with full review cycles). Experiments are run with a state-of-the-art LLM available at the time of execution; specifically, we used OpenAI's O3 model throughout.

### System message (REVIEW GUIDELINES):

You are a simulated reviewer for ICLR 2025
Here are the review guidelines you must adhere to
Review Guidelines
Below is a description of the questions you will be asked on the review form for each paper and some guidelines on what to consider when answering these questions Remember that answering no to some questions is typically not grounds for rejection When writing your review please keep in mind that after decisions have been made reviews and metareviews of accepted papers and optedin rejected papers will be made public

Summary Briefly summarize the paper and its contributions This is not the place to critique the paper the authors should generally agree with a wellwritten summary This is also not the place to paste the abstractplease provide the summary in your own understanding after reading
Strengths and Weaknesses Please provide a thorough assessment of the strengths and weaknesses of the paper A good mental framing for strengths and weaknesses is to think of reasons you might accept or reject the paper Please touch on the following dimensions
Quality Is the submission technically sound Are claims well supported eg by theoretical analysis or experimental results Are the methods used appropriate Is this a complete piece of work or work in progress Are the authors careful and honest about evaluating both the strengths and weaknesses of their work
Clarity Is the submission clearly written Is it well organized If not please make constructive suggestions for improving its clarity Does it adequately inform the reader Note that a superbly written paper provides enough information for an expert reader to reproduce its results
Significance Are the results impactful for the community Are others researchers or practitioners likely to use the ideas or build on them Does the submission address a difficult task in a better way than previous work Does it advance our understandingknowledge on the topic in a demonstrable way Does it provide unique data unique conclusions about existing data or a unique theoretical or experimental approach
Originality Does the work provide new insights deepen understanding or highlight important properties of existing methods Is it clear how this work differs from previous contributions with relevant citations provided Does the work introduce novel tasks or methods that advance the field Does this work offer a novel combination of existing techniques and is the reasoning behind this combination wellarticulated As the questions above indicates originality does not necessarily require introducing an entirely new method Rather a work that provides novel insights by evaluating existing methods or demonstrates improved efficiency fairness etc is also equally valuable

You can incorporate Markdown and LaTeX into your review
Quality Based on what you discussed in Strengths and Weaknesses please assign the paper a numerical rating on the following scale to indicate the quality of the work
4 excellent
3 good
2 fair
1 poor
Clarity Based on what you discussed in Strengths and Weaknesses please assign the paper a numerical rating on the following scale to indicate the clarity of the paper
4 excellent
3 good
2 fair
1 poor
Significance Based on what you discussed in Strengths and Weaknesses please assign the paper a numerical rating on the following scale to indicate the significance of the paper
4 excellent
3 good
2 fair
1 poor
Originality Based on what you discussed in Strengths and Weaknesses please assign the paper a numerical rating on the following scale to indicate the originality of the paper
4 excellent
3 good
2 fair
1 poor
Questions Please list up and carefully describe questions and suggestions for the authors which should focus on key points ideally around 35 that are actionable with clear guidance Think of the things where a response from the author can change your opinion clarify a confusion or address a limitation You are strongly encouraged to state the clear criteria under which your evaluation score could increase or decrease This can be very important for a productive rebuttal and discussion phase with the authors
Limitations Have the authors adequately addressed the limitations and potential negative societal impact of their work If so simply leave yes if not please include constructive suggestions for improvement In general authors should be rewarded rather than punished for being up front about the limitations of their work and any potential negative societal impact You are encouraged to think through whether any critical points are missing and provide these as feedback for the authors
Overall Please provide an overall score for this submission Choices
10 Strong Accept Technically flawless paper with groundbreaking impact on one or more areas of AI with exceptionally strong evaluation reproducibility and resources and no unaddressed ethical considerations

9 Accept Technically solid paper with high impact on at least one subarea of AI or moderatetohigh impact on multiple areas with goodtoexcellent evaluation resources and reproducibility

8 Weak Accept Technically sound and reasonably wellevaluated but not outstanding Impact may be more narrow or incremental

7 Borderline Accept Reasons to accept slightly outweigh reasons to reject A solid paper with some limitations eg ablation clarity scope

6 Weak Reject Reasons to reject slightly outweigh reasons to accept Paper may lack novelty strong results or thorough evaluation

5 Reject Paper has clear weaknesses in either technical content clarity originality or evaluation

4 Strong Reject Paper suffers from significant issues in methodology experimental support or contribution

3 Serious Reject Major flaws in correctness relevance or ethical considerations

2 Severe Reject Paper makes minimal scientific contribution or is fundamentally flawed

1 Desk Reject Level Not suitable for review due to major violations eg formatting plagiarism or entirely out of scope
Confidence Please provide a confidence score for your assessment of this submission to indicate how confident you are in your evaluation Choices
5 You are absolutely certain about your assessment You are very familiar with the related work and checked the mathother details carefully
4 You are confident in your assessment but not absolutely certain It is unlikely but not impossible that you did not understand some parts of the submission or that you are unfamiliar with some pieces of related work
3 You are fairly confident in your assessment It is possible that you did not understand some parts of the submission or that you are unfamiliar with some pieces of related work Mathother details were not carefully checked
2 You are willing to defend your assessment but it is quite likely that you did not understand the central parts of the submission or that you are unfamiliar with some pieces of related work Mathother details were not carefully checked
1 Your assessment is an educated guess The submission is not in your area or the submission was difficult to understand Mathother details were not carefully checked

**System message (LLM as a judge):**

You are an expert NLP evaluator. Given the two textual statements below:

Generated statement:
{ predicted }

Real statement:
{ real }

Please answer the following questions:

1. How many distinct points were raised in the *Real statement*? Make a checklist.
2. Now, please now count how many of the generated points overlap with those in the checklist and return the scalar number.

Return your answer in the following format:
Real points: <number> – [<point1>, <point2>, ...]
Hits (overlap): <number>

## B.2. A.1 Details for Experiment 1: Review Component Generation.

### Input Data per Instance:

- Paper Title

- Paper Abstract

- Paper Content in text

- If predicting rebuttals, paper initial review

### Few-Shot ICL Prompt Structure:

{REVIEW_GUIDELINES}
You are an expert reviewer. Given a few example papers' title, abstract {f", and its {paper_content_desc}" if paper_content else ""}, and real review comments, infer the key strengths of the current test input paper in your own words.

––– EXAMPLE 1 –––
PAPER TITLE: [Example Paper 1 Title]
PAPER ABSTRACT: [Example Paper 1 Abstract]
PAPER CONTENT: [Example Paper 1 Content]

POTENTIAL STRENGTHS:
– [Strength 1 from human review of Ex1]
– [Strength 2 from human review of Ex1]
POTENTIAL WEAKNESSES:
– [Weakness 1 from human review of Ex1]
– [Weakness 2 from human review of Ex1]
( if rebuttal )
INITIAL SCORE: [Actual initial score for Ex1]
INITIAL REVIEW: [Actual review for Ex1]
AUTHOR REBUTTAL: [Actual author rebuttals to Ex1]

––– EXAMPLE 2 –––
PAPER TITLE: [Example Paper 2 Title]
PAPER ABSTRACT: [Example Paper 2 Abstract]
PAPER CONTENT: [Example Paper 2 Content]

POTENTIAL STRENGTHS:
– [Strength 1 from human review of Ex2]
– [Strength 2 from human review of Ex2]
POTENTIAL WEAKNESSES:
– [Weakness 1 from human review of Ex2]
– [Weakness 2 from human review of Ex2]
( if rebuttal )
INITIAL SCORE: [Actual initial score for Ex2]
INITIAL REVIEW: [Actual review for Ex2]
AUTHOR REBUTTAL: [Actual author rebuttals to Ex2]

––– ACTUAL TASK –––
PAPER TITLE: [Test Paper Title ]
PAPER ABSTRACT: [Test Paper Abstract]
PAPER CONTENT: [Test Paper Content]

POTENTIAL STRENGTHS:
[LLM generates this ]
POTENTIAL WEAKNESSES:
[LLM generates this ]
( if rebuttal )
INITIAL SCORE: [Actual initial score ]
INITIAL REVIEW: [Actual review]
AUTHOR REBUTTAL: [LLM generates this]

### Notes on Evaluation:

- LLM-as-Judge for semantic coverage: Prompt another powerful LLM (the "judge") with the human review points and the AI-generated points. Ask the judge to determine, for each human point, if it was semantically captured by the AI. Calculate precision (AI points that are valid) and recall (human points captured by AI).

## B.3. A.2 Details for Experiment 2: Rating Prediction.

**Input Data per Instance:**

- Paper Title

- Paper Abstract

- Paper Content in text

- Initial Review

- Initial Review Score (for the specific review)

- Author Rebuttal

**Few-Shot ICL Prompt Structure:**

```
REVIEW_GUIDELINES
You are an expert reviewer. Based on a paper's {paper_content_desc}, an initial review, and the author rebuttal, predict what the final score (rating) should be after considering the rebuttal. Return only a single
     number.

––– EXAMPLE 1 –––
PAPER TITLE: [Example Paper 1 Title]
PAPER ABSTRACT: [Example Paper 1 Abstract]
PAPER CONTENT: [Example Paper 1 Content]
INITIAL SCORE: [Actual initial score for Ex1]
INITIAL REVIEW: [Actual review for Ex1]
AUTHOR REBUTTAL: [Actual author rebuttals to Ex1]
FINAL_RATING: [Actual final rating for Ex1]

––– EXAMPLE 2 –––
PAPER TITLE: [Example Paper 2 Title]
PAPER ABSTRACT: [Example Paper 2 Abstract]
PAPER CONTENT: [Example Paper 2 Content]
INITIAL SCORE: [Actual initial score for Ex2]
INITIAL REVIEW: [Actual review for Ex2]
AUTHOR REBUTTAL: [Actual author rebuttals to Ex2]
FINAL_RATING: [Actual final rating for Ex2]

––– ACTUAL TASK –––
PAPER TITLE: [Test Paper Title]
PAPER ABSTRACT: [Test Paper Abstract]
PAPER CONTENT: [Test Paper Content]
INITIAL SCORE: [Actual initial score for Test Paper]
INITIAL REVIEW: [Actual review for Test Paper]
AUTHOR REBUTTAL: [Actual author rebuttals to Test Paper]
FINAL_RATING:
[LLM generates this]
```

### Notes on Evaluation:

- Predicting exact final scores is hard. More informative might be predicting the *direction* of change (increase, decrease, no change) and the *magnitude* of change if any (e.g., +/- 1 point, +/- 2 points).

- This task critically highlights the need for data where reviewers *explicitly state why their score changed (or not) based on the rebuttal*. Without this, the LLM is learning from coarse signals.

## B.4. A.3 Details for Experiment 3: AI-Assisted Generation of "Reviewer Report Card" Feedback

**Input Data per Instance:**

- Paper Abstract (to provide context for the review's relevance)

- Full text of one Human-Written Review

**Few-Shot ICL Prompt Structure:**

```
You are an AI assistant tasked with providing constructive feedback on a peer review to help the reviewer improve.
Based on the paper's abstract and the provided review, generate a "Reviewer Report Card" commenting on:
1. Coverage: Did the review address key aspects like novelty, significance, technical soundness, and empirical validation relative to the abstract?
2. Specificity: Were the critiques concrete and actionable? Were praises specific?
3. Constructiveness: Was the feedback framed to help authors improve?
4. Tone: Was the language professional and respectful?

––– EXAMPLE 1 –––
PAPER ABSTRACT: [Abstract of Example Paper 1]
HUMAN REVIEW TEXT: [Full text of a human review for Example Paper 1 – ideally a review with known good/bad points]
```

REVIEWER REPORT CARD:
− Coverage: Good coverage of methodology and experiments. Novelty assessment could be more detailed.
− Specificity: Weakness point \#2 ("needs more experiments") is vague. Suggestion: "Specify which experiments would strengthen the claims, e.g., on dataset X or against baseline Y."
− Constructiveness: Overall constructive.
− Tone: Professional.

−−− EXAMPLE 2 −−−
PAPER ABSTRACT: [Abstract of Example Paper 2]
HUMAN REVIEW TEXT: [Full text of another human review for Example Paper 2]

REVIEWER REPORT CARD:
− Coverage: Focused heavily on minor presentation issues, but missed a potential flaw in the main theoretical claim mentioned in the abstract.
− Specificity: Commendable specificity in pointing out typos.
− Constructiveness: Suggestions for improvement are limited.
− Tone: Slightly dismissive in the summary ("Obvious incremental work."). Suggestion: "Rephrase to focus on specific technical reasons for limited novelty."

−−− ACTUAL TASK −−−
PAPER ABSTRACT: [Abstract of Test Paper]
HUMAN REVIEW TEXT: [Full text of Test Human Review]

REVIEWER REPORT CARD:
[LLM generates this feedback across the four dimensions]

## Notes on Evaluation:

- The "Expert-Authored Feedback on Review" for the few-shot examples would be the hardest to source. Initially, these might need to be carefully crafted by the researchers to exemplify good feedback.

- Evaluation would be primarily qualitative, involving experienced reviewers or ACs rating the LLM's feedback on dimensions like: Accuracy (does the LLM correctly identify strengths/weaknesses of the review?), Helpfulness (would this feedback help the original reviewer improve?), Actionability (are the suggestions concrete?).

- This aligns with the spirit of the ICLR 2025 Review Feedback Agent, which provided actionable suggestions to reviewers.

These more detailed conceptual setups underscore that while ICL can provide initial insights, the development of robust, reliable AI assistants for peer review will necessitate dedicated datasets, fine-tuning, and sophisticated human-in-the-loop evaluation methodologies. The primary purpose of these illustrative experiments in the position paper is to argue for the *potential* and to highlight *what is needed* to realize it.

