# OpenReview forum: "Position: The ML Community Must Build an AI-Augmented Peer-Review Ecosystem"
_ICML.cc/2026/Position_Paper_Track — ICML 2026 Position Paper Track spotlight_

### Official Review · Reviewer_SURT · 2026-03-11

**Significance:** 4
**Argument Clarity:** 4
**Rating:** 4
**Confidence:** 4

**Questions:**

Why do you constraint the work specifically to the ML field? Peer review scaling is a relevant challenge in other fields as well and our solutions *should* translate to address their needs. Brief discussion on this would be useful. Are there some things that would need to be done differently in other fields where e.g. the reviewers may have more limited technical skills? Can we pilot platforms that could be productized for use in other fields? How would the situation change if we are talking about a journal review process where revisions are prepared during the process? Should we use AI also in fields where the submission numbers are sufficiently low and there are plenty of reviewing resources?

**Alternative Views Section:**

No

**Compliance With Llm Reviewing Policy A Conservative:**

Affirmed.

**Discussion Potential:**

4

**Final Justification:**

I am a bit torn with this paper. On one hand it is a well-crafted position taking a stance on a timely challenge. On the other hand, it dismisses many of the actual challenges and proposes a rather technology-driven solution for a complex problem that is currently actively debated across several scientific fields. It is critically important for our field, as the only field that can provide technical solutions, to have active discussion on the topic, but I would like to see the discussion spearheaded by papers that take a more holistic perspective to the question. In the end, this position paper rather directly proposes a technical solution without adequately addressing the limitations or alternative perspectives.

I was seriously disappointed by the generality and tone of the author responses that were carefully crafted to seemingly address my concerns and to signal agreement with my concerns, but that in reality provided very little substance and tried to hide all subjective opinions the authors may have on the actual matter. My impression based on the rebuttal was that the authors do not take the concerns seriously. Due to this I lowered my rating slightly, but overall I am still mildly leaning towards acceptance. I do trust the authors can be a bit more transparent with the alternative views and possible objections others (within out field but also more broadly) could have on the position, and with such additions this paper does contribute to the discussion well with a thorough overview on the possible technical solutions and their challenges.

**Paper Summary:**

The authors take part in the extremely timely and active discussion on peer review in the current context where submissions to ML conferences are growing exponentially and increasingly more capable AI tools are available. The position is that we should have research and infrastructure on AI-assisted peer review, aiming for LLM as collaborative agents for various actors. The authors explain, for instance, technical challenges that would need to be addressed to create the infrastructure and discuss the need for data to train the agents.

**Position:**

Yes

**Position In Title:**

No

**Related Work:**

3

**Strengths And Weaknesses:**

The current situation is very clearly communicated for the reader and the topic is of high relevance for everyone in the ML community. As a topic this is highly suitable for a position paper and likely to result in active discussion with strong opinions. The authors' position is clear, arguing in favour of actively embedding LLMs within the review process but in a controlled manner, but the submission formatting does not satisfy the evaluation criteria: The word "position" is missing from the title, and the vague phrase "AI Imperative" does not communicate the actual position that we need to use AI (in specific ways).

The authors cover the ML literature, both from the perspective of the current challenges and from the perspective of LLM development, extremely well. However, broader literature on peer review and the scientific process is much more limited and one could argue that the position is not properly grounded on what e.g. philosophy of science literature says about the role of the review process. The emphasis is justified for a position paper in a ML venue proposing technical research agenda for the ML field, but the paper would be stronger if it was connected also to the broader perspective of what peer review and AI tools means for science in general. The perspective is also a bit narrow, starting from the assumption that we will be keeping the review process otherwise identical, with individual reviewers, ACs and rebuttal rounds, missing an opportunity to also discuss how the review process itself might need to be changed. It is perfectly okay to have a sharp focus, but some references and brief discussion on other possible dimensions involved would not hurt.

The discussion on possible roles of AI is comprehensive and again grounded well to the literature. The different aspects are communicated well and the narrative is supported by good use of structure. The illustrative example is useful, though the paper would probably worked well even without that.

The main weakness is the limited discussion in Alternative Perspectives. In effect, the only alternative perspective is "no use of AI at all" and the authors do not open the argumentation of previous works (Ye et al. 2024) but only state their position. Other possible positions, such as more limited AI usage (e.g. reviewers and authors using LLMs, but ACs not), fully automated review process done using LLMs satisfying the needs outlined in this work, changes in the review protocol itself (e.g. post-publication reviews), or substantial reduction of submissions that enter human inspection (that would be done by more senior researchers, with to without AI), are not even mentioned. These are not strawmen but proposals that have been presented in various contexts.

**Support:**

3

---

> ### Author Rebuttal · Authors · 2026-03-30
>
> We thank the reviewer for your thoughtful and candid review. We especially appreciate that you distinguished between agreement with the paper's importance and the paper's current degree of Position Track compliance. Your comments are exactly the kind that improve a position paper, and we agree with the central issues you identified.
>
> ### 1. On title compliance and stating the position clearly
>
> You are right that the current title is not fully compliant with the ICML 2026 Position Paper CFP, which requires that the title begin with `Position:` and clearly state the position. We have fixed this directly in the camera ready revision. Our revised title:
>
> `Position: The ML Community Must Build an AI-Augmented Peer-Review Ecosystem`
>
> This states the thesis explicitly rather than relying on the more rhetorical phrase "AI Imperative."
>
> ### 2. On the missing strength of the current `Alternative Views` discussion
>
> We also agree that the paper should do more than defend itself against a single anti-AI stance. In the camera ready revision, we have created a standalone `Alternative Views` section in the main body and have expanded it to engage multiple credible alternatives, including:
>
> - AI should be restricted to clerical or retrieval-oriented support only, with all evaluative synthesis reserved for humans;
> - AI for authors/reviewers but not ACs or decision support;
> - stronger process redesign rather than workflow augmentation (e.g., senior triage, post-publication review, fewer papers entering full human review);
> - a more automation-forward position in which sufficiently strong models eventually replace large parts of review.
>
> Short-form of some of the text we have added:
>
> "A serious alternative is not only 'no AI in peer review,' but also 'less AI, used in fewer places,' or 'change the review protocol rather than augment the current one.' For example, one might prefer stronger editorial triage, senior-reviewer filtering, or post-publication review over instrumenting today's conference workflow. Our position is that these protocol reforms are important to discuss, but they do not remove the immediate need for human-centered AI assistance in ML, where scale, reviewer scarcity, and existing digital infrastructure make augmentation the most actionable near-term path."
>
> ### 3. On why this is framed as an ML position rather than a science-wide one
>
> We agree that the paper should answer this more directly. Our intended claim is `ML-first, not ML-only`. We focus on ML because the pressure is most acute here: submission growth is extreme, the field already uses digital platforms such as OpenReview, code/artifact culture makes grounding and reproducibility support especially tractable, and the community is uniquely capable of building the very AI systems being discussed. But the agenda is transferable.
>
> Short-form text we have added:
>
> "We frame this as an ML position because ML is currently the most urgent and instrumentable setting for AI-augmented peer review, not because the underlying problem is unique to ML. In lower-volume or differently organized fields, the same toolkit may be optional rather than imperative, and the balance between automation, human expertise, and policy constraints may differ."
>
> ### 4. On journals, broader science, and process reform
>
> For journal workflows with multiple revision rounds, richer longitudinal traces become even more valuable because one can track how specific critiques are resolved across iterations. For fields with lower submission pressure or more limited technical reviewer fluency, the strongest use cases may shift toward author support, factual verification, and editorial triage rather than intensive reviewer-facing copilots. We have added a brief broader-science framing paragraph in the camera ready revision to make clear that our argument is about a family of sociotechnical design choices, not a one-size-fits-all ML export.
>
> ### 5. On the broader perspective and literature of peer review and science
>
> We agree that the paper can better connect ML-specific concerns to the broader functions of peer review in science: validation, deliberation, quality control, and incentive shaping. We have added broader context in the related works and discussion sections, so that the paper reads less as "keep the current pipeline and add LLMs" and more as "within a broader design space, here is the ML-specific position we defend."
>
> ---
>
> *Your review identified the two most important revisions we should make: fully complying with the Position Track format, and engaging a broader set of credible alternative positions. We are grateful for that, and we have revised the manuscript accordingly; we hope these improvements address your concerns and that you would consider updating your score. We remain happy to engage in further discussions.*

---

> > ### Author Rebuttal · Reviewer_SURT · 2026-04-02
> >
> > I think the new title is more appropriate summary of the position.
> >
> > I would naturally be happy to see the alternative views perspectives listed here discussed in more detail, but I am rather worried about your response that looks almost like a LLM-response to the request, exactly summarizing only my rather ad-hoc list of alternative views in a tone designed to please the reviewer. My list was by no means comprehensive and may have included aspects that are not real alternative views, and your response gives the impression you are not taking the concern as seriously as you should. The paragraph you wrote is disconnected from the literature, listing hypothetical positions and not grounding them on concrete examples or previous work. For example, post-publication peer review is an established concept with notable scientific literature and you would need a proper discussion on whether the scaling issues plague also that solution, whether the same AI-supported platform would be equally useful in that etc.
> >
> > Similarly, "the balance between automation, human expertise, and policy constraints may differ" is a really lazy way of discussing the potential later expansion to other fields. It is sufficient for framing the expansion to be out of scope of this paper, but otherwise does not tell the reader anything substantial.
> >
> > Regarding the literature on broader perspective: You refer to having made the chances to the camera ready, but do not provide any examples of the literature you would cover. I would like to see a concrete example, since this is again not a triviality that is resolved by adding a few references and say that the focus is in ML. If you argue you want to change part of the scientific process in a fundamental way, you should use argumentation that leverages the scientific literature on the scientific process.
> >
> > Overall, I still think the paper covers the technical perspective well but my concerns relating to grounding the research on what we already know about the form and importance of peer review were emphasized by the response. I am considering lowering my grade since I believe this is massively important. We as a ML community have the responsibility to do this right and present the proposals so that they can be accepted by the rest of the scientific community, instead of delivering technical solutions in isolation.

---

### Official Review · Reviewer_GTvj · 2026-03-12

**Significance:** 4
**Argument Clarity:** 4
**Rating:** 5
**Confidence:** 4

**Questions:**

None.

**Alternative Views Section:**

Yes

**Compliance With Llm Reviewing Policy A Conservative:**

Affirmed.

**Discussion Potential:**

3

**Final Justification:**

This position paper tackles an important problem and makes a well-supported case for its position, covering both an interesting proposal of how LLMs should be used in peer review (not just to provide reviews, but instead to accelerate and improve the work of all participants) and what the remaining challenges are.  The author rebuttal was helpful, only not altering my score because my score was already high.

**Paper Summary:**

This paper presents a position that the AI community needs to develop and embrace new technologies for AI-assisted peer review that assist authors, peer reviewers, and ACs in their participation in the peer review process.  It presents a few experiments suggesting the promise and current gaps of technology for the task, and highlights the need for the community to gather fine-grained interaction data.

**Position:**

Yes

**Position In Title:**

Yes

**Related Work:**

3

**Strengths And Weaknesses:**

Strengths
This position paper tackles an important problem and makes a well-supported case for its position.  The intro opens with a variety of relevant facts to motivate its position (the deluge of submissions, limitations in existing peer review decision quality, and promising signs for LLM assistance in peer review).  Along the way, the paper makes a solid case for peer review as an interesting AI challenge task.

I expect this paper to tell me what AI will handle, and what humans will continue to do, and what the interface between the two needs to look like.  Figure 3 illustrates this, and the paradigm here seemed creative to me; rather than for example simply introducing LLM reviewers, the approach is instead to have LLMs offering feedback at various stages.  The approach includes assistance to authors (checking against sources, code review), reviewers (reviewer report cards), and ACs.  The paper goes through each form of assistance in detail, and provides references lending credence to the plausibility of its recommendations.

I would also expect this paper to tell me what the unsolved challenges are, and what research remains to tackle those.  One primary proposed objective is for the community to collect and release fine-grained activity traces of human peer review, which can be used for AI training and evaluation—this position ends up being the most significant actionable first step I took away from the paper, and I’d suggest the authors emphasize it more explicitly in the introduction.  Another identified gap is detecting weaknesses, not just strengths in papers (Table 1).  A further objective (powered by the first) is developing new models and tools for assisting with peer review that keep humans in control.


Weaknesses

Of the enhancements proposed here, the ones for use by ACs seem helpful but also are standard things that ACs are likely already doing---in cases where the venue allows it.
“This confines AI to surface-level pattern matching.”----I feel like AI is not constrained in this way, although it may lack some of the richest problem-specific data, it can still bring to bear all of the reasoning capabilities of modern models.

I think it would be worthwhile to acknowledge some works that have called out technical limitations that automated reviewers suffer from today.  E.g. the below works show that automated reviewers lack specificity and struggle to identify contradictions in papers, respectively (although these are just a couple of probably many possible examples).

MARG: Multi-Agent Review Generation for Scientific Papers, D’Arcy et al. 2024.

Automatic Reviewers Fail to Detect Faulty Reasoning in Research Papers: A New Counterfactual Evaluation Framework, Dycke and Gurevych, 2025.

The Related Work section is a bit terse and covers some not-very-related work (e.g. SciBERT and SPECTER are important for scholarly document processing but there is much more closely related work published on automated peer review per se).

Finally, Figure 1 is comically small.  I would cut what content is necessary to make the figures legible.

**Support:**

4

---

> ### Author Rebuttal · Authors · 2026-03-30
>
> We thank the reviewer for your strong and incisive review. We especially appreciate that you engaged the paper at the level we hoped for: not "should there be LLM reviewers?", but "what should AI handle, what should humans retain, and what infrastructure is missing?" Your comments substantially improve how we position the paper.
>
> ### 1. On making the richer-data thesis more central
>
> We agree completely that the most actionable takeaway should be foregrounded earlier: the immediate bottleneck is not only model capability, but the lack of fine-grained, structured traces of human peer-review reasoning. In the camera-ready revision, we have moved this claim forward in the Introduction and have made it one of the paper's first explicit contributions.
>
> Short-form text we have added:
>
> "The key near-term bottleneck is not merely better models, but better process data: score-change rationales, rebuttal-to-judgment links, and structured traces of reviewer and AC deliberation. Without these, AI systems can imitate the surface form of reviews more easily than they can learn the reasoning process that makes peer review valuable."
>
> ### 2. On AC assistance being more than summarization
>
> We agree that, if phrased too narrowly, AC support can sound like automating what good ACs already do. We have revised this section to make clear that the value is not replacing AC synthesis, but scaling and standardizing high-friction parts of it: conflict detection across reviews, identifying unaddressed concerns after rebuttal, highlighting unusually weak or unsupported reviews, linking score changes to stated rationales, and drafting structured meta-review scaffolds that remain under AC control.
>
> Short-form text we have added:
>
> "For ACs, the point is not merely summary generation. The stronger opportunity is structured decision support: surfacing conflicts, tracing whether rebuttal responses resolved specific objections, highlighting unusually weak or unsupported reviews, linking score changes to stated rationales, and creating auditable meta-review scaffolds that help ACs deliberate more consistently while retaining full authority over the final decision."
>
> ### 3. On replacing the `surface-level pattern matching` wording
>
> We agree. The phrase 'surface-level pattern matching' is too broad; the more precise challenge is that current datasets lack the explicit supervision needed for training and evaluation. Because the 'why' behind a reviewer's decision is often unobserved, even strong models are forced to imitate outcomes without learning the reasoning process behind them.
>
> Short-form text we have added:
>
> "With current public datasets, the limitation is less generic reasoning capability than missing supervision: the data rarely records why a reviewer changed a score, which rebuttal point resolved an objection, or how ACs weighed conflicting evidence. As a result, models are pushed toward outcome imitation rather than learning auditable judgment processes."
>
> ### 4. On closer related work in automated peer review
>
> We agree that the related work section should prioritize papers with more closely related automated-reviewing approaches over generic scholarly-document models.
>
> We now include a discussion of these papers within our related work:
> - D'Arcy et al., 2024 [1] show that structured, multi-agent review generation can improve specificity and helpfulness, reducing generic comments and increasing good comments. This strengthens our claim that targeted assistive structure matters.
> - Dycke and Gurevych, 2025 [2] show that current automatic reviewers exhibit no significant response to faulty research logic, and explicitly recommend skill-specific evaluation plus human-LLM collaboration. This strongly supports our human-in-the-loop position and our argument that full automation is not yet a defensible endpoint.
>
> Actions taken: we have extended our related works to include review-generation literature such as `ReviewerGPT`, `AgentReview`, `Automated peer reviewing in paper sea`, D'Arcy et al., 2024, Dycke and Gurevych, 2025 ahead of generic scientific-document models, and have used them to sharpen the distinction between promising assistance and unreliable full automation.
>
> ### 5. On Figure 1 readability
>
> We have enlarged Figure 1 so it is legible in the main paper.
>
> ---
>
> *Your review improved the paper in three concrete ways: it pushed us to foreground the richer-data thesis, sharpen the AC-support section from "summary" to "structured decision support," and update the related work around the most relevant automatic-reviewing literature. We are grateful for this, and we have revised the manuscript accordingly; we hope your concerns have been addressed and that you would consider updating your score; we remain happy to engage in further discussions.*

---

> > ### Author Rebuttal · Reviewer_GTvj · 2026-04-01
> >
> > Thank you for the thoughtful response.

---

### Official Review · Reviewer_mAj1 · 2026-03-13

**Significance:** 3
**Argument Clarity:** 3
**Rating:** 4
**Confidence:** 4

**Questions:**

- I suggest adding a separate section entitled Alternative Views, rather than a mixed section "Alternative Perspectives and Challenges".
- I suggest adjusting the line types in the legend of Figure 2 to be consistent with those in the figure (Reviewer/Author Words Count).

**Alternative Views Section:**

Yes

**Compliance With Llm Reviewing Policy A Conservative:**

Affirmed.

**Discussion Potential:**

3

**Final Justification:**

good paper

**Paper Summary:**

This paper argues that the rapid growth of submissions to major ML conferences such as NeurIPS, ICML, and ICLR is creating a scalability crisis in peer review, where relying solely on human reviewers makes it increasingly difficult to maintain review quality, consistency, and timeliness. The authors therefore propose AI-assisted peer review as an urgent research and infrastructure direction.
Importantly, the paper does not advocate replacing human reviewers. Instead, it proposes a human-centered, AI-augmented review ecosystem in which LLMs assist reviewers with fact checking, related-work search, review feedback, and consistency calibration; assist authors with improving clarity, identifying weaknesses, and preparing rebuttals; and assist area chairs with summarizing feedback, identifying disagreements, and drafting meta-reviews. The authors argue that the key bottleneck for building such systems is not only model capability but also the lack of fine-grained, structured peer-review process data, such as score-change rationales and reviewer–author interaction traces.
The paper also presents illustrative experiments on ICLR data. Results show that LLMs can partially extract strengths and rebuttal points but remain limited in identifying weaknesses and predicting score changes. These findings support the authors’ claim that current public data is insufficient for developing stronger AI-assisted peer-review systems.

**Position:**

Yes

**Position In Title:**

Yes

**Related Work:**

3

**Strengths And Weaknesses:**

Strengths
- The topic is interesting and of significant discussion value for the peer review.
- The submission provides sufficient and solid evidence to convincingly verify that AI assistance for peer review is an urgent necessity.
- It reports the results of illustrative experiments, which makes the work more credible compared with other paper.

Weaknesses
-  Figure 1 is presented in a small size, likely due to page limitations, which reduces readability.
- The discussion of Alternative Views is less comprehensive than that of other position papers in my batch.

**Support:**

3

---

> ### Author Rebuttal · Authors · 2026-03-30
>
> We thank the reviewer for your constructive and carefully targeted feedback. We especially appreciate that you found the core position timely and well supported, while also identifying exactly where the paper can better satisfy the ICML Position Track criteria in presentation and structure. Your comments are actionable, and we have incorporated them directly.
>
> ### 1. On `Alternative Views` versus the current mixed section
>
> We agree with your main point: the current `Alternative Perspectives and Challenges` section should be split, and the alternative positions should be made more explicit and more comprehensive. In the camera-ready revision, we have replaced it with a standalone main-body section titled `Alternative Views`, separate from challenges/limitations.
>
> More importantly, we have expanded it beyond the single "no AI" alternative. The revised section explicitly discusses four credible opposing positions:
>
> - AI should be restricted to clerical or retrieval-oriented support only, with all evaluative synthesis reserved for humans;
> - AI may assist authors/reviewers, but not ACs or final decision support;
> - the better solution is to redesign peer review itself (e.g., stronger triage, senior-reviewer models, post-publication review), not augment the current pipeline;
> - Fully automated reviewing may be a long-term aspiration, but it is not reliable enough today for scientific judgment.
>
> Short-form text we have added:
>
> "A credible alternative view is not that AI should never appear in peer review, but that it should remain confined to clerical or retrieval-oriented support, with all evaluative synthesis reserved for humans. Another is that the community should reform review protocol itself, for example, through stronger triage or post-publication review, rather than instrumenting the present workflow. We take these alternatives seriously; our position is that, given current submission scale and current model limitations, the most defensible near-term path is human-centered augmentation with humans retaining final judgment."
>
> ### 2. On Figure 1 readability
>
> We agree that Figure 1 is currently too small. This is a presentation issue, not a conceptual one, and we have fixed it. In the camera-ready version, we have enlarged the figure footprint and reclaimed space from nearby text/formatting so that the axes, density shape, and trend are legible without zooming.
>
> ### 3. On Figure 2 legend consistency
>
> We agree on the legend mismatch. We have revised Figure 2 so that the legend line styles exactly match the corresponding plotted series for reviewer and author word-count traces. This is a straightforward but worthwhile correction, and thank you for catching it.
>
> ### 4. On the manuscript-level improvements prompted by your review
>
> Your review helped us see that the paper should not only argue a strong position, but also make its alternatives visibly discussable in the format ICML requested. Accordingly, we have:
>
> - created a standalone `Alternative Views` section;
> - expanded that section to engage multiple credible alternatives;
> - improved the readability/consistency of Figures 1 and 2.
>
> ---
>
> *We appreciate the specificity of your suggestions. They not only improve the presentation; they sharpen the paper's compliance with the Position Track and make the manuscript more useful as a catalyst for discussion, which is exactly the goal of this track. We have revised the paper accordingly; we hope these changes address your concerns and that you might consider updating your score. We would be happy to engage in any further discussions.*

---

> > ### Author Rebuttal · Reviewer_mAj1 · 2026-04-02
> >
> > none

---

### Official Review · Reviewer_sVnB · 2026-03-13

**Significance:** 4
**Argument Clarity:** 4
**Rating:** 6
**Confidence:** 4

**Questions:**

None

**Alternative Views Section:**

Yes

**Compliance With Llm Reviewing Policy A Conservative:**

Affirmed.

**Discussion Potential:**

4

**Final Justification:**

Copy-pasting the rebuttal acknowledgment text:

I did not have any explicit questions in my initial review but had also flagged that there was the "Position in Title" requirement wasn't currently satisfied. The authors have addressed in their rebuttal to Reviewer SURT that they will amend the title to "Position: The ML Community Must Build an AI-Augmented Peer-Review Ecosystem" to satisfy the requirements. I would like to maintain my support for the paper.

**Paper Summary:**

The paper strongly advocates for the use of LLMs to assist authors, reviewers, and Area Chairs across all the stages of the peer-review process. It tackles each stage separately and comprehensively discusses how LLMs can be used to assist the participants in those stages. It discusses that the current available datasets are insufficient to develop such assistance tools. And the ML research community as a whole needs more transparency and openness in releasing data regarding review decisions and meta-review creation. More generally, it advocates for not only using LLMs for improving the review process but also improving the review process to make it more favorable to export data for better training of future generations of such tools. For example, when a reviewer decides to change their score, they could label certain points of the provided rebuttal as the "key causal contributors" to their change in judgment. Essentially, they would like the entire peer review process to act like a well-defined data-creation (and labeling) platform to enable creation of high-quality peer-review datasets.

**Position:**

Yes

**Position In Title:**

No

**Related Work:**

4

**Strengths And Weaknesses:**

Strengths:

Well-supported: The paper does a thorough job of covering the various peer-review aspects that could be improved by LLM-assistance. They also support their position and the need for their position pretty strongly.

Relevance: They clearly argue why the topic is of relevance for the ICML community.

Discussion: Yes, the paper is likely to inspire discussion. There is already a lot of discussion about this topic in the ML community but this paper puts a lot of arguments and thoughts in a cohesive manner making it easy for other members of the community to discuss and also act on actualizing their vision.

Clarity: Yes, the position is clearly argued.

Related work: There is an extensive discussion of related work (both in the main paper and also in the Appendix). One caveat is that I am not an expert on this topic and so I cannot authoritatively state that they have covered most of the relevant related work.

**Support:**

4

---

> ### Author Rebuttal · Authors · 2026-03-30
>
> We thank the reviewer for your exceptionally thoughtful and supportive review. We were especially pleased that you identified what we also believe is the paper's most actionable contribution: not only advocating AI assistance for peer review, but treating the review process itself as a source of structured supervision for future AI systems. Your framing helped us see that this point should be made more explicitly and earlier in the paper.
>
> ### 1. On peer review as a structured data-creation platform
>
> Your summary that the workflow itself should become a better "data-creation (and labeling) platform" is exactly right, and we agree that this idea should be elevated from an implication to an explicit central claim. In the camera-ready version, we have strengthened the Introduction and the call-to-action to state more directly that our position is not merely "use LLMs in peer review," but "redesign parts of peer review so that the human reasoning already produced in review can be captured in structured, privacy-preserving form and used to build better human-centered tools."
>
> Short-form text we have added to the Introduction:
>
> "Our position is not only that AI should assist peer review, but that the peer-review workflow should be instrumented to capture richer causal traces of scientific judgment, such as why a score changed, which rebuttal sentence resolved which concern, and which unresolved issue ultimately drove the decision. Without such structured traces, even strong models are forced to imitate outcomes without learning the reasoning process behind them."
>
> ### 2. On causal rationales for score changes and rebuttal impact
>
> We particularly appreciate your concrete example that when a reviewer changes a score, they could identify the "key causal contributors" in the rebuttal or discussion. We agree. This is precisely the kind of lightweight, high-value annotation that could convert today's largely unstructured discussion logs into training and evaluation data for future reviewer, author, and AC assistants.
>
> Action taken: We have sharpened Section 5 to make this operational rather than aspirational. In particular, we have foregrounded active elicitation interfaces such as:
>
> - a rationalization prompt when a score changes;
> - explicit linking from rebuttal sentences to updated reviewer judgments;
> - structured tags for clarification, concession, unresolved objection, and empirical resolution.
>
> Short-form text we have added:
>
> "A practical first step is to add low-friction rationalization prompts at key decision points. For example: 'You changed your score from 5 to 7; which rebuttal sentence or discussion point most influenced this update, and why?' Such prompts preserve human control while turning latent judgment shifts into explicit supervision."
>
> ### 3. On the broader manuscript improvements prompted by your review
>
> Although you raised no objections, your reading helped us identify where the paper can become sharper:
>
> - making the richer-data thesis more central in the Introduction rather than introducing it mainly later;
> - making the workflow-instrumentation idea more concrete through explicit interface examples;
> - clarifying that the goal is human-centered augmentation, not automation for its own sake.
>
> ---
>
> *We are grateful for the accurate summary of the manuscript's intended contribution. Your review helped us recognize that the most distinctive part of the paper is not only the list of possible AI tools, but the argument that the community should deliberately build the data and interfaces needed for those tools to become reliable, auditable, and genuinely useful. We have revised the manuscript accordingly to materially strengthen its clarity and impact; consequently, we hope these changes address your concerns. We remain happy to engage in further discussions.*

---

> > ### Author Rebuttal · Reviewer_sVnB · 2026-04-04
> >
> > I did not have any explicit questions in my initial review but had also flagged that there was the "Position in Title" requirement wasn't currently satisfied. The authors have addressed in their rebuttal to Reviewer SURT that they will amend the title to "Position: The ML Community Must Build an AI-Augmented Peer-Review Ecosystem" to satisfy the requirements. I would like to maintain my support for the paper.

---

### Decision · Program_Chairs · 2026-04-30

**Decision:**

Accept (spotlight)

**Comment:**

Some group had to submit this position because everyone is thinking about the issue.  It is sufficiently important that it should probably done by a large consortium of senior community member as a number of working groups.
The general position is that of AI augmenting various stages of the reviewing process.
Some of the proposals are for things people are trying to arrange already, when the venue allows it.
One reviewer suggests more coverage of methods that didn't work and negative results.  Another reviewer suggests looking more broadly at the reviewing process and changing it.